# Social aversive generalization learning sharpens the tuning of visuocortical neurons to facial identity cues

Yannik Stegmann[1]*, Lea Ahrens[1], Paul Pauli[1,2], Andreas Keil[3], Matthias J Wieser[1,4]

[1]Department of Psychology (Biological Psychology, Clinical Psychology, and Psychotherapy), University of Würzburg, Würzburg, Germany; [2]Center for Mental Health, Medical Faculty, University of Würzburg, Würzburg, Germany; [3]Center for the Study of Emotion and Attention, University of Florida, Gainesville, United States; [4]Department of Psychology, Education, and Child Studies, Erasmus University Rotterdam, Rotterdam, Netherlands

**Abstract** Defensive system activation promotes heightened perception of threat signals, and excessive attention to threat signals has been discussed as a contributory factor in the etiology of anxiety disorders. However, a mechanistic account of attentional modulation during fear-relevant processes, especially during fear generalization remains elusive. To test the hypothesis that social fear generalization prompts sharpened tuning in the visuocortical representation of social threat cues, 67 healthy participants underwent differential fear conditioning, followed by a generalization test in which participants viewed faces varying in similarity with the threat-associated face. We found that generalization of social threat sharpens visuocortical tuning of social threat cues, whereas ratings of fearfulness showed generalization, linearly decreasing with decreasing similarity to the threat-associated face. Moreover, individuals who reported greater anxiety in social situations also showed heightened sharpened tuning of visuocortical neurons to facial identity cues, indicating the behavioral relevance of visuocortical tuning during generalization learning.

*For correspondence:
yannik.stegmann@uni-wuerzburg.de

Competing interests: The authors declare that no competing interests exist.

## Introduction

Selectively perceiving and differentially responding to cues associated with threat versus safety is a fundamental function of the vertebrate brain. The dysregulation of these functions is at the core of many psychiatric problems. Over the past decade, basic and applied researchers interested in mental health have focused on the contribution of dysfunctional associative learning mechanisms to the etiology of anxiety disorders (*Dymond et al., 2015*). Given its intuitive relation with exaggerated fear and anxiety, the process of overgeneralization—showing threat responses to safety cues that resemble threat-associated stimuli—has been of particular interest. However, previous clinical and translational work has yielded contradictory findings. While some authors observed overgeneralization in patients with anxiety disorders (*Kaczkurkin et al., 2017*; *Lissek et al., 2014b*; *Lissek et al., 2010*), others did not (*Ahrens et al., 2016*; *Tinoco-González et al., 2015*). This lack of convergent findings may be due to the fact that different physiological systems respond differently to varying similarity with a fear stimulus.

When individuals are in a state of fear, defensive mechanisms are activated with the goal of engaging in adaptive action, for example in fighting or escaping the threat. This defensive engagement is indexed by somato-visceral measures, such as fear-potentiated startle, skin conductance, and cardiovascular responses (*Boecker and Pauli, 2019*; *Bradley and Lang, 2000*). These measures have been considered in previous fear generalization experiments (*Ahrens et al., 2016*; *Lissek et al., 2010*; *Torrents-Rodas et al., 2013*), with mixed results regarding the nature and

variability of fear generalization across a range of cues varying in similarity with a threat cue (**Ahrens et al., 2016**). In addition to preparing autonomic and motor efferent systems for action, defensive mobilization also includes heightened sensory processing, that is, perception and attention to threat (**Robinson et al., 2013**). In line with this notion, a substantial body of research has shown that stimuli predicting threat are attended more than neutral cues, and that heightened attention toward threatening stimuli is pronounced in patients with anxiety disorders (**Bar-Haim et al., 2007**). At the same time, several studies focused on the importance of perceptual discriminability of threatening stimuli (**Struyf et al., 2017**; **Zaman et al., 2019**). As a consequence, excessive attention to threatening stimuli has been discussed as causal or contributory in the etiology of anxiety disorders (**Clark and Wells, 1995**; **Rapee and Heimberg, 1997**). Direct neurophysiological evidence of heightened attention to threat in clinical disorders is scarce, however, and findings of both heightened and diminished attention have been reported. The absence of a mechanistic account and lack of direct unequivocal evidence of attention dysfunction may be a result of the using indirect measures of attention to threat. The present study uses electrophysiological measurements from visual cortex to test mechanistic hypotheses derived from the structure and function of the human visual system.

Direct visuocortical responses to a specific stimulus may be quantified with the steady-state visually evoked potential (ssVEP, **Müller et al., 1998**). The ssVEP is an oscillatory neuronal response to stimuli that are periodically modulated in luminance. Heightened ssVEP amplitudes mark increased visuocortical activation and can be used to index attentional processes. For example, attended features (**Müller et al., 2006**) and selective spatial attention (**Müller et al., 1998**) facilitate visuocortical activation compared to unattended features and locations. SsVEPs are also sensitive to emotional processes and show increased amplitudes for emotional compared to neutral stimuli (**Keil et al., 2003**; **Kemp et al., 2002**; **McTeague et al., 2011**). Therefore, they provide a promising method for testing hypotheses regarding changing perception and attention as participants undergo fear generalization learning (**Wieser et al., 2016**). The amplitude of the ssVEP differentiates threat from safety signals, being selectively heightened for conditioned threat cues (reviewed in **Miskovic and Keil, 2012**). Building on these findings, **McTeague et al., 2015** utilized ssVEPs to study population-level tuning of orientation-selective neurons in the primary visual cortex during fear generalization. In the neuroscience literature, tuning functions are often used to to describe differences in sensitivity of a response (neural or behavioral) along a physical feature gradient. For example, orientation tuning functions denote how neurons in the retinotopic visual cortex selectively respond to specific orientations (see **Figure 1**). At the population level, especially with scalp record fields, information regarding the tuning of individual neurons is obscure. Changes in the preferential tuning of population-level responses along a feature gradient can however be assessed with suitable research designs: To examine the extent to which aversive learning affects the population-level orientation tuning reflected in ssVEPs, **McTeague et al., 2015** used high-contrast grating stimuli differing in orientation as conditioned stimuli. During pre-

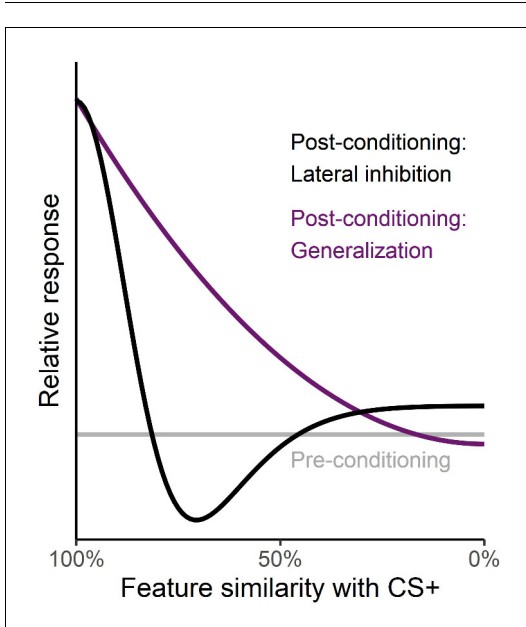

**Figure 1.** Different tuning functions during aversive learning. The flat grey line depicts relative (behavioral or neural) responses along a physical feature gradient during pre-conditioning. The black curve shows a possible tuning function for lateral inhibition after aversive conditioning as seen in orientation-selective neuronal populations in the visual cortex (**McTeague et al., 2015**). Relative responses are enhanced for the specific feature associated with the aversive event and supressed for the most similar features. In contrast, the purple curve depicts a gradually decreasing generalization gradient observed in self-report and somato-visceral indices of generalization learning.

conditioning, the ssVEP amplitude evoked by each orientation was the same, resulting in a flat tuning curve. During acquisition, only the grating stimulus in the middle of this stimulus continuum was paired with an aversive noise (i.e. the CS+). After several conditioning trials, the CS+ prompted enhanced visuocortical engagement, accompanied by a suppression of the grating orientations with highest similarity to the CS+. This tuning pattern, which contrasts with the gradually decreasing generalization gradient observed in self-report and somato-visceral indices of learning, suggests lateral inhibitory interactions among orientation-selective neuronal populations in the visual cortex.

In the present study, we tested the hypothesis that sensory systems, when presented with a similarity gradient around a social threat stimulus, undergo changes to sharpen their tuning properties toward the relevant feature. Paralleling work on orientation-selective neurons discussed above, we expected amplification of visuocortical responses to the threat-associated face and a selective suppression of responses to the face most similar to the threat-associated face, reflecting inhibitory interactions between neuronal populations that represent facial features. This hypothesis is grounded in work showing substantial evidence for single-unit and population level (LFPs, fMRI) tuning in face-specific areas in the human and primate brain (*Freiwald and Tsao, 2010*; *Freiwald et al., 2009*; *Gilaie-Dotan and Malach, 2007*; *Leopold et al., 2006*; *Loffler et al., 2005*). These studies have demonstrated that there are neurons and neuronal populations in face-sensitive cortical areas, like the occipital face area (OFA) and the fusiform face area (FFA), which show gradual responses to varying facial identify, often referred to as 'tuning' to facial identities (*Chang and Tsao, 2017*). Here, we examine the malleability of population-level tuning as observers learn to associate one identity along a gradient of morphs with an aversive outcome. To further establish the behavioral relevance of visuocortical tuning, we also examine the extent to which such sharpened visuocortical tuning is associated with interindividual differences in social anxiety.

## Results

### Habituation
#### Steady-state visually evoked potentials (ssVEPs)
To induce ssVEPs, two different facial stimuli (CS+ and CS-) were presented with a flickering frequency of 12 Hz (see *Figure 2*). After converting the electrocortical signal to current source density (CSD) estimates and transforming it into the frequency-domain, the signal-to-noise ratio (SNR) was calculated by dividing the power of the driving frequency by the mean of the spectral power at six adjacent frequency bins, leaving out the two immediate neighbors. The resulting SNRs were pooled across eight sensor locations over the occipital pole for statistical analyses. During habituation, the linear mixed model analysis including the within-factor CS-type (two levels: CS+ vs CS-) and social anxiety as a continuous between-variable revealed a significant main effect for social anxiety, indicating that higher social anxiety was associated with higher ssVEP-SNRs in general and a non-significant trend for CS-type (CS+: $M = 3.61$, $SD = 1.91$; CS-: $M = 3.39$ $SD = 1.77$). There was no CS-type x social anxiety interaction (see *Table 1*).

#### Valence and arousal ratings
The linear mixed model analysis of valence ratings revealed neither a significant main effect of social anxiety nor a CS-type x social anxiety interaction (see *Table 1*). However, subjects rated the CS- as slightly more unpleasant than the CS+, (CS+: $M = 4.60$, $SD = 1.22$; CS-: $M = 4.99$ $SD = 1.25$). With regard to arousal ratings, there was no significant effect for CS-type (CS+: $M = 3.69$, $SD = 1.61$; CS-: $M = 3.58$ $SD = 1.58$), social anxiety, or CS-type x social anxiety interaction. This result points out that none of the two faces elicited more arousal at the beginning of the experiment.

### Acquisition
#### SsVEPs
After the CS+ had been paired with the US (see *Figure 2*), linear mixed model analyses for ssVEPs yielded a main effect of CS-type and a small effect of social anxiety, demonstrating that subjects reacted with higher amplitudes to the CS+ ($M = 3.69$, $SD = 1.80$) compared to the CS- ($M = 3.33$, $SD = 1.56$, see *Figure 3a*) and that higher social anxiety was associated with higher amplitudes in general (see *Table 2*). There was no significant CS-type x social anxiety interaction.

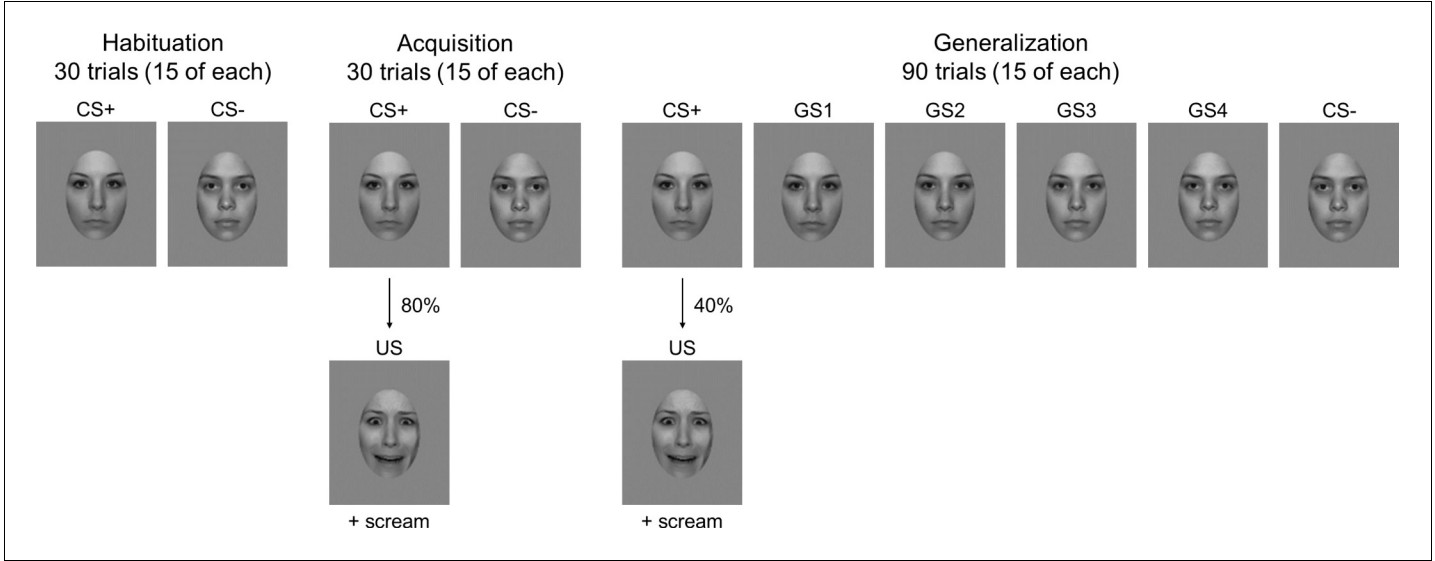

**Figure 2.** Experimental design. Habituation, acquisition and generalization phase are illustrated. Stimuli were randomly presented for 3 s during each of the three parts of the experiment. The US consisted of a 1500 ms presentation of the respective CS+ face displaying a fearful expression, which was accompanied by a 95 dB shrill female scream. The assignment of face to CS+/CS- was counterbalanced between participants.

## Valence and arousal ratings

After acquisition, there was a significant main effect of CS-type for both valence and arousal ratings (see *Table 2*), demonstrating that subjects rated the CS+ as more unpleasant ($M$ = 6.42, $SD$ = 1.41) and more arousing ($M$ = 6.51, $SD$ = 1.63) compared to the CS- (unpleasantness: $M$ = 4.73, $SD$ = 1.34; arousal: $M$ = 3.69, $SD$ = 1.94, see *Figure 3c and d*). Main effects of social anxiety and CS-type x social anxiety interactions were not significant.

## US expectancy rating

The analysis detected a main effect of CS-type (see *Table 2*), with higher US expectancy ratings for the CS+ ($M$ = 72.09, $SD$ = 21.71) compared to the CS- ($M$ = 8.66, $SD$ = 18.74), underlining that the experimental manipulation was effective (see *Figure 3b*). There was no effect of social anxiety or CS-type x social anxiety interaction.

**Table 1.** Results of the linear mixed model analyses during habituation.

**ssVEP-SNRs:**

| | | | | |
|---|---|---|---|---|
| CS-Type | $F(1,65)$=3.18 | p=0.079 | $R^2$=0.047 | $CI$ = [.000,. 187] |
| Social anxiety | $t(65)$=2.18 | p=0.033 | $\beta$ = 0.48 | $SE$ = 0.21 |
| CS-Type x Social anxiety | $F(1,65)$=0.69 | p=0.408 | $R^2$=0.011 | $CI$ = [.000,. 112] |
| **Valence:** | | | | |
| CS-Type | $F(1,65)$=4.07 | p=0.048 | $R^2$=0.059 | $CI$ = [.001,. 207] |
| Social anxiety | $t(65)$=1.34 | p=0.186 | $\beta$ = −0.15 | $SE$ = 0.12 |
| CS-Type x Social anxiety | $F(1,65)$=1.16 | p=0.285 | $R^2$=0.018 | $CI$ = [.000,. 130] |
| **Arousal:** | | | | |
| CS-Type | $F(1,65)$=0.34 | p=0.562 | $R^2$=0.005 | $CI$ = [.000,. 096] |
| Social anxiety | $t(65)$=1.24 | p=0.221 | $\beta$ = 0.21 | $SE$ = 0.17 |
| CS-Type x Social anxiety | $F(1,65)$=2.18 | p=0.145 | $R^2$=0.032 | $CI$ = [.000,. 162] |

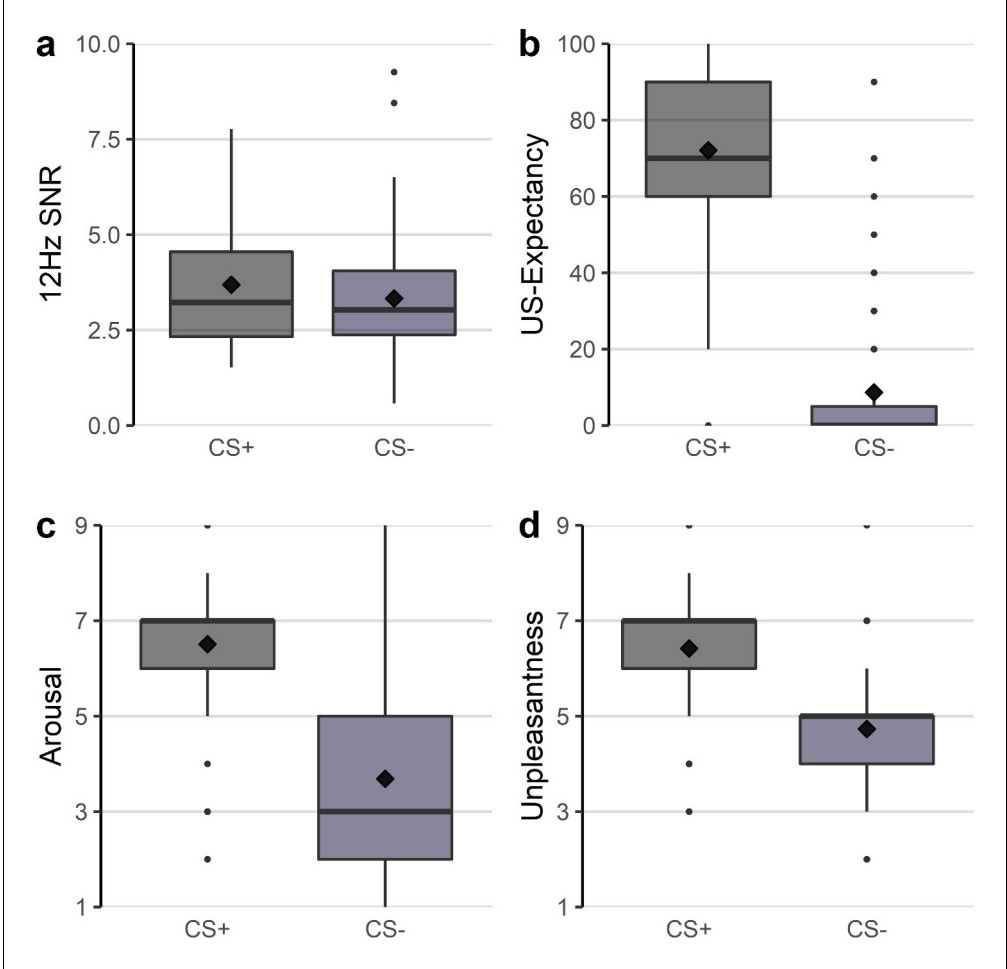

**Figure 3.** Boxplots and means (diamonds) of the (**a**) 12 Hz ssVEP signal-to-noise ratios (SNR) during the acquisition phase and mean US expectancy (**b**), arousal (**c**), and unpleasantness ratings (**d**) after acquisition.

## Generalization

### Steady-state visually evoked potentials

During the generalization test, four additional faces (GS1 – 4) were presented by morphing the two faces of the previous phases together in 20% steps, with the GS1 being the most similar to the CS+ and the GS4 being the most similar to the CS- (see *Figure 2*). The CS-type (six levels: CS+ *vs* GS1 vs GS2 vs GS3 vs GS4 *vs* CS-) x social anxiety linear mixed model provided a main effect of CS-type and a marginal effect of social anxiety (see *Figures 4* and *5a* and *Table 3*). Post-hoc contrasts indicated significant differences between GS4 *vs* CS-, $t(325) = 2.94$, p=0.003, GS3 *vs* CS-, $t(325) = 3.24$, p=0.001, and CS+ *vs* CS-, $t(325) = 2.17$, p=0.031, but not between GS2 *vs* CS-, $t(325) = 1.75$, p=0.081, and GS1 *vs* CS-, $t(325) = 0.82$, p=0.411. However, the omnibus-test revealed no stimulus type x social anxiety interaction.

To follow up on the frequentist statistics and to seek converging evidence from a Bayesian perspective, we compared the fit of a lateral inhibition pattern to a quadratic and linear trend, using weighted predictors in Bayesian linear models (see *Figure 6*). To this end, we constructed weight vectors that reflected the predictions of the alternative hypotheses for the experimental conditions. This approach allowed us to leverage the specific model predictions under the lateral inhibition, generalization, and linear hypotheses, and to quantify the fit between the empirical data and model predictions in one test across all conditions. The lateral inhibition pattern was expressed as the difference of two Gaussians (weights: +2,–2, +0.5, +1, +0.5,–2 for CS+, GS1, GS2, GS3, GS4, CS-), paralleling previous studies on visuocortical tuning (*Antov et al., 2020*; *McTeague et al., 2015*).

**Table 2.** Results of the linear mixed model analyses during acquisition learning.

**ssVEP-SNRs:**

| | | | | |
|---|---|---|---|---|
| CS-Type | $F(1,65)=5.50$ | $p=0.022$ | $R^2=0.078$ | $CI = [.003,. 235]$ |
| Social anxiety | $t(65)=2.00$ | $p=0.050$ | $\beta = 0.37$ | $SE = 0.19$ |
| CS-Type x Social anxiety | $F(1,65)=1.57$ | $p=0.168$ | $R^2=0.013$ | $CI = [.000,. 118]$ |
| **Valence:** | | | | |
| CS-Type | $F(1,65)=49.51$ | $p<0.001$ | $R^2=0.432$ | $CI = [.271,. 587]$ |
| Social anxiety | $t(65)=0.80$ | $p=0.424$ | $\beta = 0.09$ | $SE = 0.12$ |
| CS-Type x Social anxiety | $F(1,65)=0.00$ | $p=0.982$ | $R^2=0.000$ | $CI = [.000,. 075]$ |
| **Arousal:** | | | | |
| CS-Type | $F(1,65)=91.13$ | $p<0.001$ | $R^2=0.584$ | $CI = [.447,. 705]$ |
| Social anxiety | $t(65)=0.05$ | $p=0.959$ | $\beta = 0.01$ | $SE = 0.16$ |
| CS-Type x Social anxiety | $F(1,65)=1.29$ | $p=0.259$ | $R^2=0.020$ | $CI = [.000,. 135]$ |
| **US expectancy:** | | | | |
| CS-Type | $F(1,65)=323.15$ | $p<0.001$ | $R^2=0.833$ | $CI = [.771,. 884]$ |
| Social anxiety | $t(65)=0.17$ | $p<0.867$ | $\beta = -0.30$ | $SE = 1.77$ |
| CS-Type x Social anxiety | $F(1,65)=0.15$ | $p=0.736$ | $R^2=0.002$ | $CI = [.000,. 083]$ |

The quadratic (weights: +2.5334, +1.0934,–0.0267, −0.8267,–1.3067, −1.4667) and linear (weights: +2.5, +1.5, +0.5,–0.5, −1.5,–2.5) trend were modeled after the analyses of the linear and quadratic component of the generalization gradient, which are commonly employed in the fear generalization literature (*Ahrens et al., 2016*; *Lissek et al., 2014a*; *Lissek et al., 2014b*). In a first step, each Bayesian model was compared to the 'random intercept only' model (null model, 0), before transitive Bayes factors were calculated to obtain the relative evidence of one model over another. These transitive Bayes factors allow a direct comparison between the competing models. For a summary of resulting Bayes factors for each candidate main effect, interaction and predictor weight model see *Table 4*. The main effect of weighted CS-type received support for the lateral inhibition pattern only, $BF_{La1/0} = 13.26$, but not for the quadratic, $BF_{Q1/0} = 0.14$, or linear trend, $BF_{Li1/0} = 0.13$. Further including a main effect of social anxiety did not lead to substantially increased support for the lateral inhibition pattern, $BF_{La2/La1} = 1.38$, the quadratic, $BF_{Q2/Q1} = 1.08$, or linear trend, $BF_{Li2/Li1} = 1.36$. The strongest evidence could be found for the full interaction model (SNR ~CS type + social anxiety + CS-type x social anxiety) and the lateral inhibition pattern, $BF_{La3/0} = 57.32$, which substantially extends the support of the main effects model, $BF_{La3/La2} = 3.13$, suggesting that the accentuation of the lateral inhibition pattern increased with higher social anxiety. By contrast, the full interaction models for the Quadratic, $BF_{Q3/0} = 0.11$, and Linear trend, $BF_{Li3/0} = 0.38$, still yielded less evidence

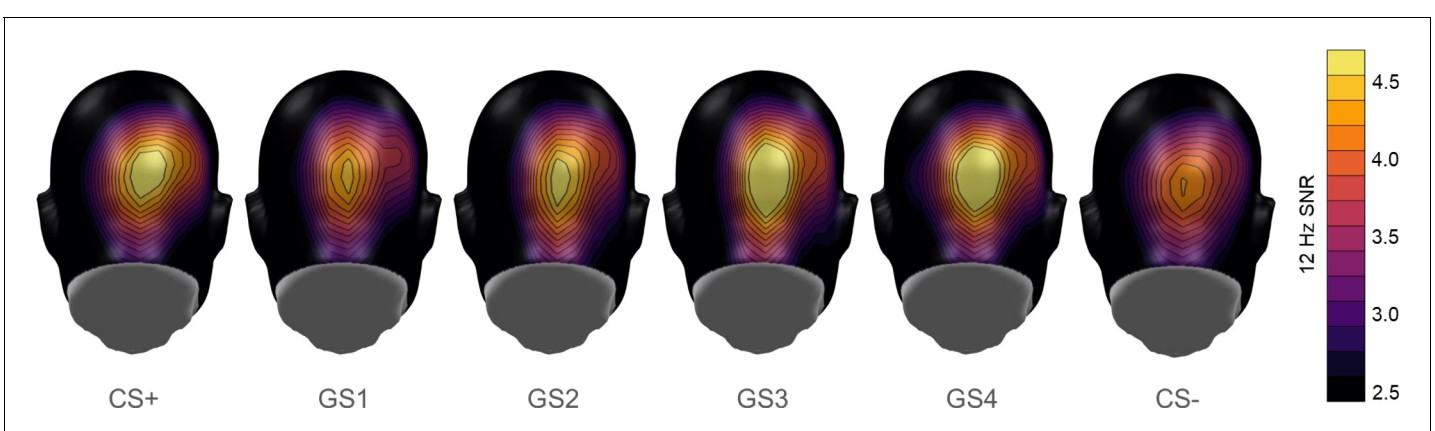

**Figure 4.** Mean scalp topographies of the 12 Hz ssVEP signal-to-noise ratios to the conditions during the generalization test.

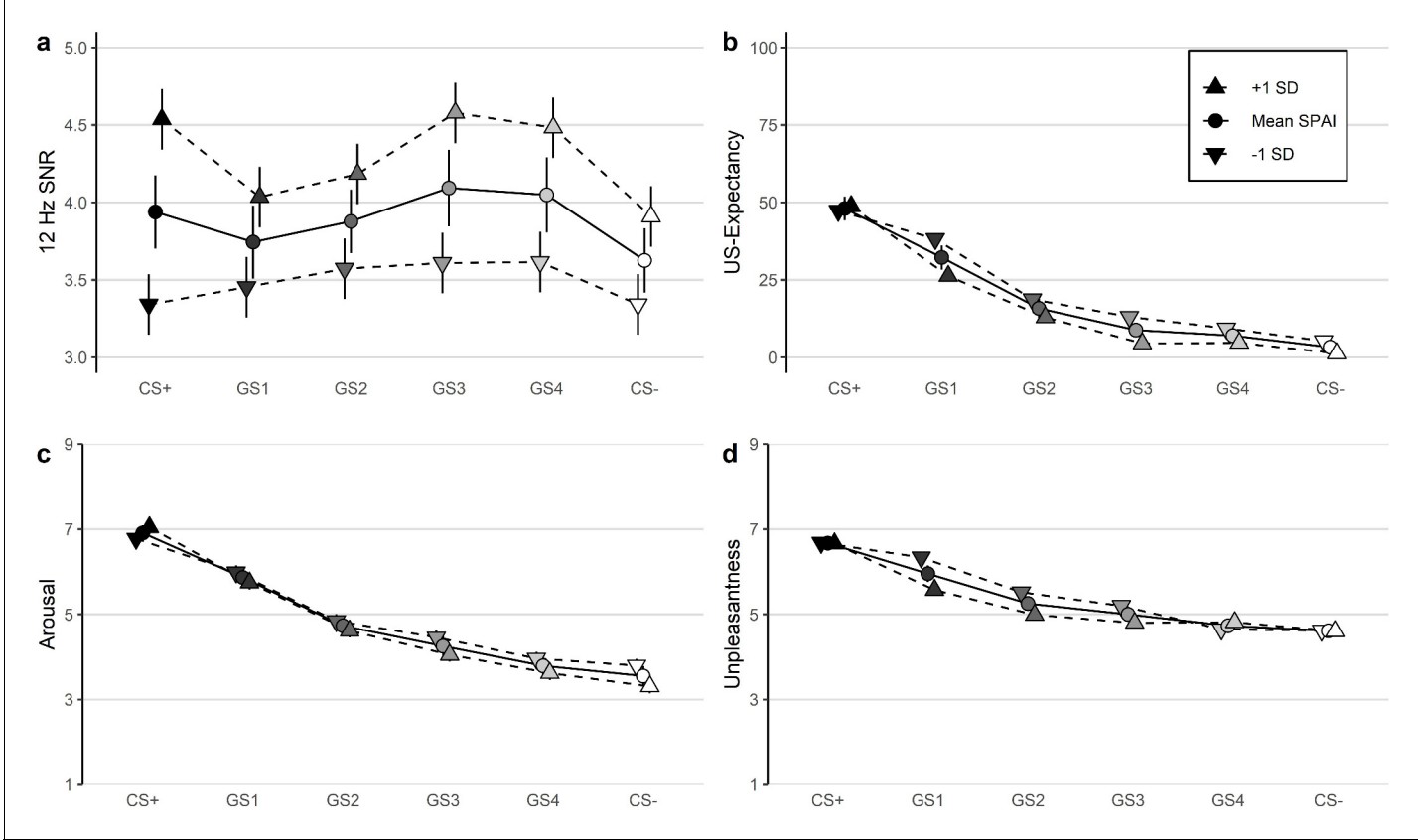

**Figure 5.** Generalization test: (**a**) Mean 12 Hz signal-to-noise ratios (SNR) ±*SEM* of the ssVEP during the generalization test. (**b**) Mean US-expectancy, (**c**) arousal and, (**d**) unpleasantness ratings ± *SEM* after generalization. Prediction intervals are shown for the *mean ±1 SD* of the SPAI covariate to illustrate the influence of social anxiety.

than the null model (see *Figure 6*). Importantly, by comparing the main and interaction effect models of each contrast against the same random intercept model, it was possible to derive the relative evidence of the lateral inhibition model over the quadratic and linear trend (see *Table 4*, last two columns). Results for the full interaction model demonstrated that the lateral inhibition model is 858 times more likely than the quadratic trend model and 502 times more likely than the linear trend model.

To follow up on the interaction of CS-type and social anxiety and to demonstrate the effect of social anxiety on the accentuation of the lateral inhibition pattern, we performed an additional regression analysis. In a first step, we calculated a visuocortical tuning index on subject level, which is the scalar product of the weights of the lateral inhibition model (2, –2, 0.5, 1, 0.5, –2) and the respective individual ssVEP responses. This visuocortical tuning index increases with a stronger accentuation of the lateral inhibition pattern and decreases if individual responses deviate from the pattern. For comparison, we calculated similar indices for the quadratic and linear trend. In a second step, we analyzed a linear regression model with social anxiety as predictor and the ssVEP indices as depended variables (see *Figure 7*). The regression analysis revealed a moderate, positive correlation between social anxiety and the visuocortical tuning index, $r(65) = .288$, p = 0.018, $BF_{1/0} = 3.65$, confirming that the accentuation of the lateral inhibition pattern increased with higher social anxiety, while the weak correlations between social anxiety and the index for the quadratic, $r(65) = .146$, p = 0.238, $BF_{2/0} = 0.52$, or linear trend, $r(65) = .137$, p = 0.268, $BF_{3/0} = 0.49$, missed significance and did not receive support from Bayesian-analyses.

**Table 3.** Results of the linear mixed model analyses during generalization learning.

**ssVEP-SNRs:**

| | | | | |
|---|---|---|---|---|
| CS-Type | $F(5,325)=3.39$ | p=0.009 | $R^2=0.045$ | $CI = [.020,. 111]$ |
| Social anxiety | $t(65)=1.94$, | p=0.056 | $\beta = 0.40$ | $SE = 0.21$ |
| CS-Type x Social anxiety | $F(5,325)=1.57$ | p=0.167 | $R^2=0.024$ | $CI = [.009,. 080$ |
| **Valence:** | | | | |
| CS-Type | $F(5,325)=35.83$ | p<0.001 | $R^2=0.355$ | $CI = [.286,. 436]$ |
| Social anxiety | $t(65)=0.17$ | p<0.867 | $\beta = 0.13$ | $SE = 0.12$ |
| CS-Type x Social anxiety | $F(5,325)=1.83$ | p=0.107 | $R^2=0.027$ | $CI = [.011,. 086]$ |
| **Arousal:** | | | | |
| CS-Type | $F(5,325)=66.80$ | p<0.001 | $R^2=0.507$ | $CI = [.443,. 574]$ |
| Social anxiety | $t(65)=1.12$, | p=0.267 | $\beta = -0.12$ | $SE = 0.18$ |
| CS-Type x Social anxiety | $F(5,325)=0.71$ | p=0.618 | $R^2=0.011$ | $CI = [.005,. 060]$ |
| **US expectancy:** | | | | |
| CS-Type | $F(5,325)=57.30$ | p<0.001 | $R^2=0.469$ | $CI = [.402,. 540]$ |
| Social anxiety | $t(65)=1.37$ | p=0.174 | $\beta = -2.79$ | $SE = 2.03$ |
| CS-Type x Social anxiety | $F(5,325)=0.97$ | p=0.435 | $R^2=0.015$ | $CI = [.006,. 067]$ |

## Valence and arousal ratings

The linear mixed models of the ratings yielded a significant main effect of CS-type for both valence and arousal ratings (see *Table 3*). Following the procedure of the ssVEP analysis, follow-up contrasts were calculated. With regard to arousal (see *Figure 5c*), subjects differentiated among the CS- and the CS+ plus three GS: CS- versus CS+, $t(325) = 14.93$, p<0.001, CS- versus GS1, $t(325) = 10.28$, p<0.001, CS- versus GS2, $t(325) = 5.24$, p<0.001, and CS- versus GS3, $t(325) = 3.12$, p=0.002, but not CS- versus GS4, $t(325) = 1.06$, p=0.290. For valence ratings (see *Figure 4d*), results showed differences between CS- and CS+, $t(325) = 10.95$, p<0.001, CS- and GS1, $t(325) = 7.14$, p<0.001, CS- and GS2, $t(325) = 3.41$, p<0.001, and CS- and GS3, $t(325) = 2.06$, p=0.040, but not between CS- and GS4, $t(325) = .64$, p=0.526. These results suggest that all subjects transferred their fear response reflected in both valence and arousal ratings from the CS+ to three GSs. In addition, there were no main effects of social anxiety nor CS-type x social anxiety interactions.

## US expectancy ratings

US expectancy analysis revealed a main effect of CS-type (see *Figure 5b* and *Table 3*). Post-hoc contrasts yielded significant effects for the comparison of CS- versus CS+ $t(325) = 13.73$, p<0.001; GS1: $t(325) = 8.88$, p<0.001; and GS2: $t(325) = 3.84$, p<0.001. The differences between CS- and GS3: $t(325) = 1.69$, p=0.091, and CS- and GS4, $t(325) = 1.14$, p=0.254, were not significant. As found in valence and arousal ratings, subjects showed an enhanced tendency to generalize their conditioned fear reaction, indicated by the fact that they expected the GS1 and GS2 to be followed by the US, although they had never been paired with the US. The main effect of social anxiety and the CS-type x social anxiety interaction were not significant.

## Discussion

The goal of the present study was to test the hypothesis that aversive generalization learning prompts sharpened representations of facial identity, reflecting inhibitory interactions between neuronal populations that represent facial features associated with threat versus safety. Second, we leveraged interindividual differences in social anxiety to examine whether the sharpened visuocortical tuning to facial identity is heightened in those characterized by higher social anxiety. For this

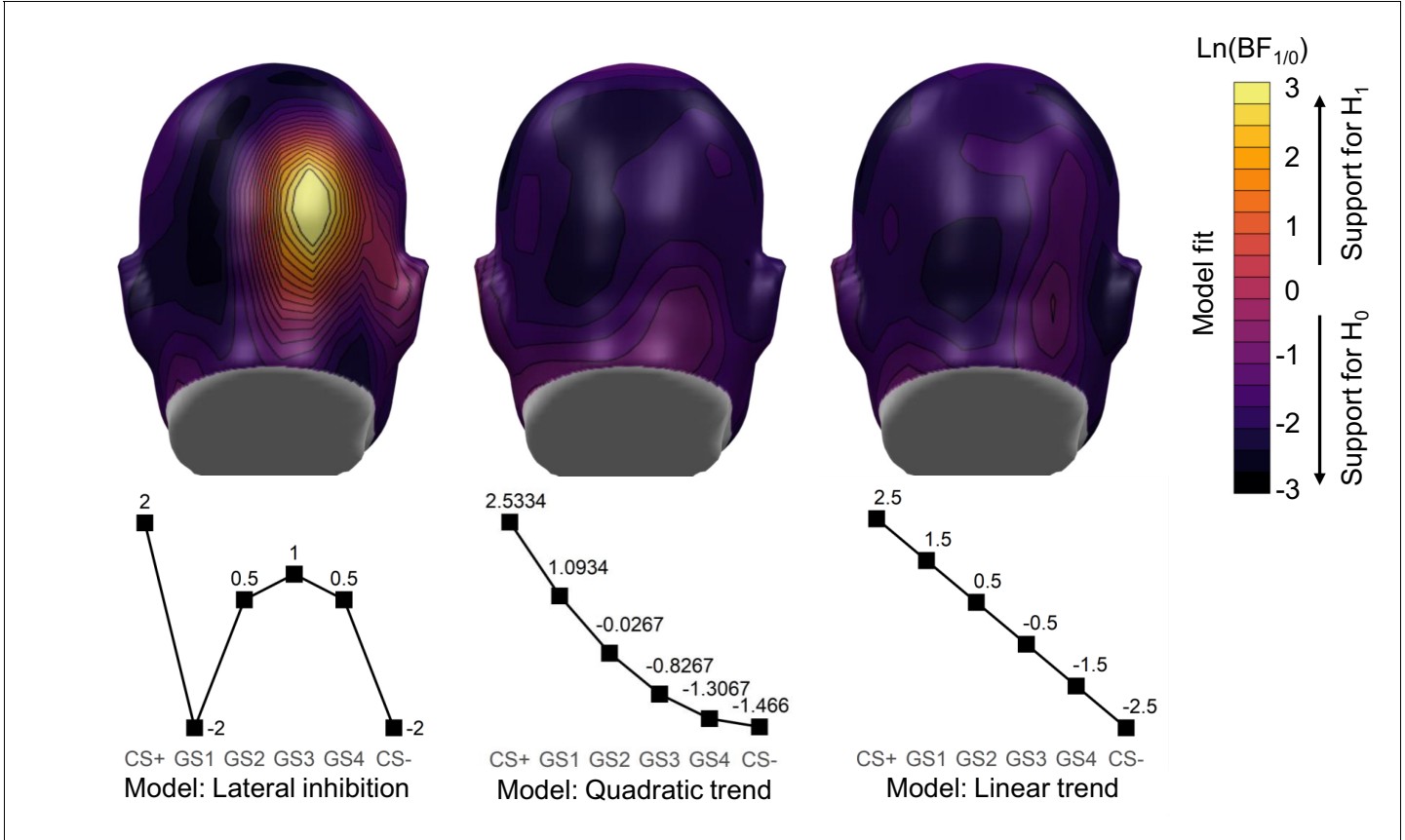

**Figure 6.** Bayesian model fit: Topographical distributions of the Bayes Factor for comparing the main effect model '*SNR ~ CS-type*' to the null model for each contrast. Weights used for the contrasts are displayed at the bottom row. Natural log-transformed BFs are illustrated, so that positive values display support for the full effect model while negative values display support for the null model.

purpose, steady-state visually evoked potentials (ssVEPs) as well as valence, arousal and US expectancy ratings were recorded in a fear conditioning and generalization paradigm with social cues.

Crucial for the later generalization test, successful fear conditioning was indexed in both ssVEP amplitude and arousal, valence and US expectancy ratings during acquisition, and social anxiety was not associated with stronger discrimination between conditioned stimuli or generally increased ratings. Thus, the CS+ during acquisition elicited increased visuocortical responses reflecting enhanced sensory engagement (*Stegmann et al., 2019a*; *Wieser et al., 2014c*). However, during habituation and acquisition, subjects with higher social anxiety showed overall amplified visuocortical responses to the face stimuli, which is in line with the notion of generally heightened sensitivity to facial expressions in social anxiety disorder (*McTeague et al., 2018*) and has been observed previously in

**Table 4.** Summary of the Bayesian linear model analysis.

| $BF_{M/0}$ | Model | Lateral inhibition | Quadratic trend | Linear trend | Inhibition vs quadratic | Inhibition vs Linear |
|---|---|---|---|---|---|---|
| $M_1$: | SNR ~ CS-type | 13.26 | 0.14 | 0.13 | 104.17 | 97.62 |
| $M_2$: | SNR ~ CStype + SA | 18.33 | 0.15 | 0.17 | 105.49 | 125.16 |
| $M_3$: | SNR ~ CS type + SA + CS-type x SA | 57.32 | 0.11 | 0.38 | 858.52 | 502.184 |

Bayes factors of main and interaction effect models ($M_1 - M_3$) compared to the 'random intercept only' model (Null model) for the lateral inhibition pattern (weights: +2,–2, +0.5, +1, +0.5,–2 for CS+, GS1, GS2, GS3, GS4, CS-), quadratic trend (weights: +2.5334, +1.0934,–0.0267, −0.8267,–1.3067, −1.4667) and linear trend (weights: +2.5, +1.5, +0.5,–0.5, −1.5,–2.5). The last two columns display direct model comparisons between the lateral inhibition pattern to the quadratic or linear trend by dividing respective BFs for each main and interaction effect model. SA, social anxiety.

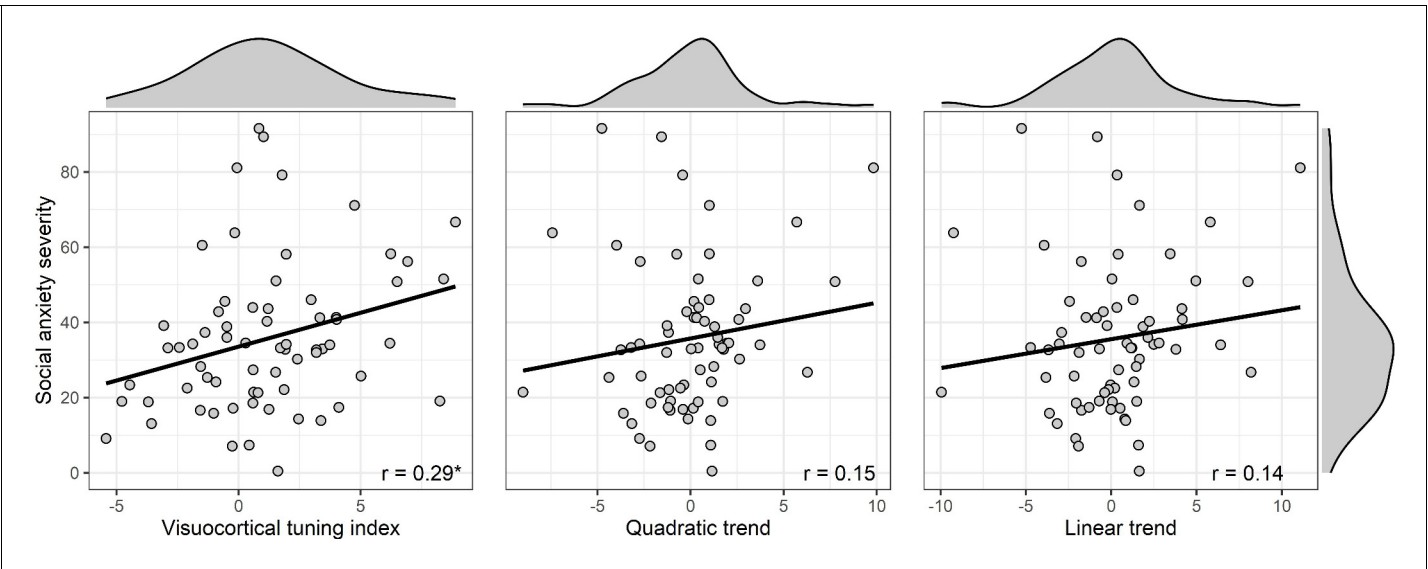

**Figure 7.** Comparison of the correlations between social anxiety and the different contrast models. Only the visuocortical tuning index as a parameter for the accentuation of the lateral inhibition pattern significantly increased with social anxiety. Marginal density plots display the distributions of the social anxiety scores and the ssVEP indices.

patients with SAD using ssVEPs (*McTeague et al., 2011*) and in high socially anxious individuals using ERP components (*Mühlberger et al., 2009*) as markers of early perceptual processing.

During fear generalization, ratings of arousal, valence and US expectancy monotonically diminished with decreasing similarity to the CS+, indicating that subjects transferred their fear response from the threat signaling face to similar faces, although those had not been associated with the aversive outcome. These results corroborate previous studies on conditioned generalization, which demonstrated gradual, monotonic, generalization effects for subjective ratings (*Lissek et al., 2008*; *McTeague et al., 2015*; *Stegmann et al., 2019b*), somato-visceral measures, such as fear-potentiated startle (*Lissek et al., 2008*), skin conductance responses (*Stegmann et al., 2019b*; *Torrents-Rodas et al., 2013*) and heart rate (*Ahrens et al., 2016*) and electrocortical responses, that is, late positive potentials (*Nelson et al., 2015*).

Importantly, the visuocortical responses did not show a monotonic generalization gradient, but instead displayed a response pattern consistent with sharpening of the threat face representation. The amplitude of the ssVEPs – in contrast to the ratings – did not gradually diminish with increasing distance from the CS+. Instead, ssVEP amplitude for the CS+ was increased, but followed by an immediate reduction for the closest GS (GS1), and a slow increase in response to the remaining GSs, before it was again reduced for the learned safety cue (CS-). This observation is consistent with the response pattern of orientation sensitive neurons in the visual cortex during a fear generalization paradigm with enhanced visuocortical engagement to the CS+ and a suppression of the grating orientations with the highest similarity to the CS+ (*McTeague et al., 2015*). In line, the Bayesian analyses demonstrated that the observed ssVEP generalization data more likely corresponds to the lateral inhibition model compared to the null model, which received more support from the Bayesian analyses than the quadratic or linear trend model.

Together, these findings suggest that there is a dissociation in the aversive generalization patterns of sensory compared to efferent and autonomic systems, which is consistent with the idea that fear generalization is an active and multifaceted process that integrates a wide variety of signals to organize adaptive fear responses (*Onat and Büchel, 2015*). A large body of work has shown that visuocortical engagement with specific stimulus features varies with the motivational relevance of these features (*Bradley et al., 2012*). The present results are in line with this notion, but also suggest that these adaptive sensory changes observed during learning differ from the efferent system's responses. To date, there are not many studies that are directly targeting the functional differences between sensory and efferent systems. Using steady-state visual evoked fields (the

magnetoencephalographic counterpart of ssVEPs), two studies could demonstrate that threat-induced sensory changes in low-level visual areas occur independently of the conscious awareness of the CS-US contingencies (*Moratti and Keil, 2009*; *Yuan et al., 2018*), suggesting that visuocortical responses are to some extent independent of higher level cognitive processes. In contrast, it has been suggested, that threat-induced changes in the sensory system represent short-term plasticity in the early visual cortex, which might be induced by projections from subcortical structures like the amygdala or thalamus (*Miskovic and Keil, 2012*). This could be one reason for the divergence between visuocortical and subjective responses, as verbal reports reflect a more cognitive component of the defensive response (*LeDoux and Pine, 2016*). On the other hand, *McTeague et al., 2015* also found a dissociation between the visuocortical and the fear-potentiated startle responses. The startle reflex, as an index of the autonomic component of the efferent system, is also assumed to be directly modulated by amygdala projections (*Davis, 2006*). In this case, the discrepancies between sensory and efferent response patterns might be mediated by different subregions of the amygdala, which modulate the sensory and efferent system according to their supposed functions in threat detection and defensive responding. Taking an evolutionary perspective, it is most adaptive for the organism to enhance sensory specificity in the visual cortex to distinguish the motivational information-providing stimulus from others sensory signals as reflected in the lateral inhibition model. On the other hand, the 'efferent' readiness to respond to a potential threat is generalized, as reflected in a monotonically decreasing generalization gradient, because a false alarm is less costly than a - potentially fatal - miss. This evolutionary interpretation assumes that the constantly changing and diverse environment in which humans find themselves demands and thus favors plastic physiological mechanisms, which may differ between systems in order to optimize functioning (*Miskovic and Keil, 2012*).

The present study also has implications regarding the neural mechanisms mediating aversive generalization learning, and regarding the neocortical changes underlying learning and memory more broadly: *McTeague et al., 2015*, using a generalization gradient of oriented gratings, explained their finding of visuocortical sharpening as a consequence of lateral inhibitory interactions among orientation-selective neuronal populations in the primary visual cortex. In this case, signals from frontoparietal attention networks may selectively facilitate CS+ representations in visual cortex, prompting local inhibitory interactions between adjacent cortical units. This process is thought to prompt suppression of the features represented by the most spatially proximal populations. In fact, ongoing computational modeling efforts in our laboratories explain these and other generalization data best by assuming that top-down signals take the shape of a generalization gradient (paralleling behavioral and autonomic data), and it is the organization of visual cortex that turns this gradient into a sharpening pattern through lateral inhibition. As such, sharpening would not be inherited by anterior areas but would result from the organization and geometry of visual cortical areas. This interpretation is further supported by research demonstrating that the orientation-tuning functions of visual neurons may be shaped by short-term plastic processes (*Dragoi et al., 2000*).

We hypothesized that in a situation in which facial identity predicts threat versus safety, a similar mechanism may operate in the cortical tissue that is specific not to orientation but to the features encoding facial identity. There are two loci where such tuning may occur: First, as aversive learning proceeds, projections from anterior structures may increasingly target lower-level representations of individual facial features in retinotopic areas, thus prompting inhibitory interactions between individual physical facial features such as orientation or spatial frequency, which differ across the morphing gradient. This is consistent with findings and models in perceptual learning (e.g. reverse hierarchy theory; *Ahissar and Hochstein, 2004*) and would prompt a mid-occipital topography of sharpening effects in the present study.

Second, inhibitory interactions may occur between similar faces along a gradient of morphs, in face-sensitive cortical areas. This alternative hypothesis is in line with recent evidence suggesting that neuronal populations in face-sensitive cortical patches encode identity by combining their population tuning to sets of high-order shape and appearance dimensions (e.g. *Chang and Tsao, 2017*). Neural populations that are sensitive to such features may be located in the early visual cortex or at later stages of the face-processing pathway, for example the occipital face area (OFA) or the fusiform face area (FFA) (*Duchaine and Yovel, 2015*). The present study suggests that the amplification of these selective neuronal cell ensembles to a given threat face prompts lateral inhibitory interactions among neurons that are selective to slightly different facial features. Our results support the

hypothesis that such interactions take place in extra-striate, higher order visuocortical areas, as we exploited the assets of high-density EEG recording and CSD-transformation leading to a precise topographical distribution of the ssVEP signal. These signal topographies reveal indeed right-lateralized activation peaks for the signal-to-noise ratios of the ssVEP signal (*Figure 4*) as well as for the model fit of the lateral inhibition pattern (*Figure 6*). Thus, our results are compatible with the idea of a right-hemispheric dominance of face perception (*Rossion, 2014*) and suggest an involvement of the right OFA or FFA in sharpening face discrimination on the basis of lateral inhibition.

Furthermore, Bayesian model and correlational analyses revealed that visuocortical system engagement is associated with self-reported symptoms of social anxiety. This result adds to a growing body of literature on attentional biases in social anxiety, including evidence from reaction-time tasks, (*Bantin et al., 2016*), eye-tracking (*Wieser et al., 2009*), event-related potentials (ERPs, *Mühlberger et al., 2009*; *Wieser et al., 2010*) and ssVEPs (*McTeague et al., 2018*; *McTeague et al., 2011*). Here, we provide further evidence that cortical engagement in response to threat-associated phobic-relevant stimuli is dimensionally related to social anxiety. It is important to mention, however, that we - replicating *Ahrens et al., 2016* – found no association between strength of social anxiety and (over)generalization of conditioned fear in terms of a higher fear responses (efferent system) to the generalization stimuli. Instead, we observed social anxiety to be associated with a more pronounced lateral inhibition pattern in visuocortical responses to generalization stimuli. Please note that the effect of social anxiety on visuocortical responses was only evident in the Bayesian-analysis but not in the corresponding ANOVA. This discrepancy results from the differences regarding statistical power between omnibus- and contrast-analysis (*Furr and Rosenthal, 2003*). The ANOVA tests for any differences, while the contrast-analysis only tests for deviations from the specified pattern. This notion is substantiated by the finding that social anxiety was not associated with visuocortical responses for the quadratic or linear trend model, but only for the lateral inhibition model. We conclude that this latter finding is indicative for the functional relevance of the lateral inhibition model in visuocortical tuning during generalization learning. Given the healthy subjects of this study, however, with those showing a psychiatric disorder being excluded, future studies should examine diagnosed and treatment-seeking SAD patients to substantiate initial findings and to draw conclusions on how sensory generalization versus sharpening contributes to the etiology or maintenance of social anxiety disorder or other psychopathologies.

One important limitation to our findings is that there was no substantial increase in differential ssVEP-SNRs (CS+ vs CS-) from habituation to acquisition, $t(66) = 0.83$, $p = 0.409$, $d = 0.10$, $CI_{95} = [-0.14, 0.34]$. A reason for the lack of such a difference may be related to the trendwise differences between CS+ and CS- at baseline, which often indicate different responses to one of the two faces. However, this seems unlikely in our study, because the mapping of the two faces to the CS+ and CS- was fully counterbalanced between subjects. In addition, in the differential conditioning literature, cross-phase comparisons are typically avoided because they confound any conditioning effects with time-dependent effects such as adaptation, in which both the CS+ and CS- evoked responses decline over time, from habituation to acquisition and finally into the extinction phase. This has been well established in meta-analyses (e.g. *Fullana et al., 2016*), and is true for a wide range of dependent measures, including startle, skin conductance, fMRI, and ssVEPs where this pattern of findings has been noted and systematically addressed several times (e.g. *Keil et al., 2013*; *Moratti and Keil, 2005*). Thus, fear conditioning studies interested in temporal changes previously utilized analysis on a trial-by-trial level in order to avoid confoundation with these types of adaptation processes (e.g. *Sjouwerman et al., 2016*; *Weike et al., 2007*; *Wieser et al., 2014c*).

Post-hoc analyses comparing pre- and post-conditioning: although a lack of statistically robust changes between the experimental phases would not affect our interpretation of the main results regarding the generalization phase, we addressed this potential concern in a post-hoc analysis quantifying the amount of conditioning-related effects above and beyond the initial difference in habituation by means of parametric bootstrapping in combination with Bayesian analyses (*Efron, 2012*).

To this end, we computed a distribution for the CS+ versus CS- difference in ssVEP amplitude separately for habituation (to be used as prior distribution) and acqusition (posterior distribution) based on 100,000 bootstrapped group mean differences. Odds for conditioning effects to occur were estimated from these two distributions (the odds of the CS+-CS- difference to be positive), and the BF of interest was given as the ratio of posterior (acquisition) over prior (habituation) odds. The error of this process was estimated by repeating the above process 100 times and measuring

the standard error of the resulting mean. This BF corresponds to the change in confidence that the ssVEP for the CS+ is greater than ssVEP for the CS- in acquisition (the posterior), relative to the prior distribution, which was estimated from the CS+ versus CS- difference during habituation. This analysis yielded a BF (posterior odds over prior odds) of 4.77, error = 1.3%, suggesting that the acquisition data provided substantial evidence for the notion that fear conditioning selectively amplifies the CS+ evoked ssVEP above and beyond the differences present during habituation.

In conclusion, our study extends current notions of generalization learning, by demonstrating the involvement of inhibitory interactions among feature-specific neurons in the visuocortical system during fear generalization to facial stimuli. We found that the accentuation of the lateral inhibition pattern increased with the severity of social anxiety. Future research may examine stability of the lateral inhibition response pattern during extinction as well as the role of perceptual sharpening in fear extinction learning.

## Materials and methods

### Subjects

Subjects were 67 undergraduate students (age: $M$ = 24.10, $SD$ = 6.33; 48 female) with normal or corrected vision and without past or present psychiatric diagnosis or family history of epilepsy (self-report), who were paid or received course credit for participation. The sample size was based on the effect sizes of previous aversive conditioning studies using ssVEPs (*McTeague et al., 2015*; *Miskovic and Keil, 2013*) and adapted for covariate-analyses. Subjects completed the Social Phobia and Anxiety Inventory (SPAI, German version, *Beidel et al., 1989*) as a self-report measurement of social anxiety ($M$ = 67.70, $SD$ = 19.74, $Min$ = 32.48, $Max$ = 126.60). Prior to participation, written informed consent was obtained from each participant. The study was approved by the ethics review board of the Medical Faculty of the University of Würzburg (87/13).

### Stimuli and apparatus

Conditioned stimuli (CS) consisted of two pictures of female actresses with a neutral facial expression taken from the NimStim Set of Facial Expressions (*Tottenham et al., 2009*). Pictures were adjusted for luminance and brightness, converted to gray-scale and presented using Presentation (Neurobehavioral Systems, Inc, Albany, CA). One of the actresses was randomly selected as threat cue for each participant (CS+) while the other served as safety signal (CS-). Pictures were shown on a gray background on a 17-inch monitor (resolution = 1,280×1,024 pixel) in a flickering mode at a frequency of 12 Hz in order to elicit ssVEPs. Face-specific areas are often targeted with relatively slow driving frequencies (*Baldauf and Desimone, 2014*), but face-specific processing has also been isolated from paradigms with faster frequencies (*Campagnoli et al., 2019*; *Gruss et al., 2012*; *Wieser et al., 2014a*; *Wieser et al., 2014c*).

The US consisted of the respective CS+ face displaying a fearful expression and a simultaneously presented 95 dB shrill female scream of the IADS (*Bradley and Lang, 1999*). Four generalization stimuli (GS) were created by morphing the two faces in 20% steps using a face-morphing software (Squirlz Morph; Xiberpix, Solihull, UK). The GS most similar to the CS+ is referred to as GS1 and the GS most similar to the CS- as GS4 (see *Figure 2*).

### Design and procedure

The experiment comprised three blocks (habituation, acquisition, generalization). Habituation and acquisition consisted of 30 trials (two faces, each presented 15 times), while there were 90 trials in the generalization phase (six faces, each presented 15 times), resulting in 150 trials in total. After completing the questionnaires, EEG electrodes were applied to participants, who were seated in a noise-reduced, darkened room one meter distant to the screen. In the habituation phase of the experiment, faces were presented for 3000 ms without reinforcement. During acquisition, one of the faces (CS+) was paired in 12 of 15 trials (80% reinforcement) with the US, which lasted 1500 ms and was presented at the offset of the CS+ with a sound volume of 95 dB by Labtech speakers (Labtech International Ltd., Ringmer, East Sussex, GB) and a Kenwood KA-3010-Amplifier (Kenwood Electronics, Heusenstamm, GER). Subjects were not informed of any specific relation among the CSs and the US prior to the experiment and the assignment of faces to CS-conditions was counterbalanced

across subjects. Generalization consisted of the CS+, CS-, and four GS, each presented 15 times (90 trials). While CS- and GS were never reinforced, 6 of the 15 CS+ were still followed by the US to prevent early extinction (40% reinforcement, *Lissek et al., 2008*) (see *Figure 2*). The presentation order of the faces within each block was pseudo-randomized such that no more than two of the same faces could occur in a row. After each trial, a gray screen with a fixation cross was presented. Inter-trial intervals differed between 2000 and 2500 ms. At the end of each phase, subjects rated the valence (ranging from 1 - 'very pleasant' to 9 - 'very unpleasant') and arousal (ranging from 1 – 'very calm' to 9 – 'very arousing') of the faces using a computer-based version of the Self-Assessment Manikin Scale (*Bradley and Lang, 1994*). Moreover, subjects were asked to rate US expectancy after acquisition and generalization from 0% to 100% as a response to the question 'What is the likelihood that the currently presented face is followed by a scream?' to measure successful learning of the CS-US association.

## EEG recording and analysis

Electrocortical activity was measured via 129 electrodes using an Electrical Geodesics (EGI, Eugene, OR) high-density EEG System referenced to Cz, recorded with a sampling rate of 250 Hz and online bandpass filtered with 0.1 and 100 Hz and a 50 Hz notch filter. The threshold of impedances was kept below 50 kΩ as recommended for the Electrical Geodesics high-impedance amplifiers. Offline, EEG analyses were implemented using the software EMEGS (Electro Magnetic Encephalography) for Matlab (*Peyk et al., 2011*). First, epochs of 600 ms pre-stimulus and 3000 ms post-stimulus onset were extracted, and data were filtered with a low-pass filter of 40 Hz. In a second step, artifact rejection was conducted according to the SCADS procedure (*Junghöfer et al., 2000*). This way, outlying channels could be identified and interpolated from the full channel set and artifact-contaminated trials could be excluded from the analyses. Trials were rejected when more than 20 channels out of 129 were outliers as per the statistical parameters used for artifact identification. In a next step, remaining trials were averaged in the time domain for each subject according to the different experimental conditions. To increase spatial resolution of the EEG signal, we then calculated the current source densities (CSD) of the time-averaged data. The CSD transformation offers the advantage of reducing the negative impact of volume conduction and thereby effectively minimizing unwanted topographical variability between subjects by quantifying source densities (*Junghöfer et al., 1997*; *Kayser and Tenke, 2015*). The CSD approach relies on the spatial Laplacian (the second spatial derivative) of the scalp potential to estimate the potential distribution at the cortical surface. Here, we used the CSD algorithm described by *Junghöfer et al., 1997*, with λ = 0.2, which is well suited for dense-array EEG montages and has been used in previous studies investigating ssVEP responses to facial stimuli (*McTeague et al., 2011*) or during fear generalization (*McTeague et al., 2015*). The CSD time series values were then transformed into the frequency domain using a Fast-Fourier-Transformation on a time interval between 500 and 3000 ms after stimulus-onset. The first 500 ms after stimulus onset were omitted due to initial non-stationary components of the ssVEP. In a next step, we obtained the signal-to-noise ratio (SNR) for the driving frequency of 12 Hz by dividing the power of the driving frequency by the mean of the spectral power at six adjacent frequency bins, leaving out the two immediate neighbors. The SNR is a unitless measure that accounts for both the evoked signal and the random noise in the data and has recently been used in other ssVEP paradigms as well (*Barry-Anwar et al., 2018*; *Boylan et al., 2019*).

For statistical analysis, the ssVEP activity was pooled across sensor Oz and seven neighbouring electrodes (EGI sensors 70, 71, 72, 74, 75, 76, 82, 83; *Wieser et al., 2014a*; *Wieser and Keil, 2014b*).

## Statistical analysis

Stimulus differences in ssVEP amplitudes as well as valence and arousal ratings during habituation and acquisition were analyzed with linear mixed models with the within-subject factor CS-type (2: CS +, CS-) and mean-centered SPAI scores as covariates. Both main effects and the interaction of CS-type and SPAI were entered as fixed effects. Subjects were entered as random intercepts to the model. For the generalization phase, the same linear mixed model was analyzed, though the factor CS-type now included six levels (CS+, GS1, GS2, GS3, GS4, CS-). Follow-up tests for significant effects of CS-type were analyzed using simple contrasts with the CS- as reference level (*Lissek et al.,*

*2008*). US expectancy ratings underwent the same analyses except for the habituation phase, because US expectancy ratings just started after the acquisition phase. Significance was evaluated using the Kenward-Rogers approximation for degrees of freedom (*Kenward and Roger, 1997*; *Luke, 2017*). Alpha was set at p<0.05 (two-tailed). Linear mixed models were conducted in the R software environment (version 3.5.0.; *R Development Core Team, 2020*), using the package 'lme4' (version 1.1–20; *Bates et al., 2015*). Standardized effect sizes and confidence intervals for the discrete factors of the linear mixed models were calculated as partial-$R^2$, using the package 'r2glmm' (version 0.1.2; *Jaeger et al., 2017*).

To compare the lateral inhibition pattern to a quadratic and linear fit, Bayesian linear models were used. For this analysis, a pre-specified weight vector for each contrast entered the linear regression as predictors. The lateral inhibition pattern was expressed as the difference of two Gaussians (weights: +2,–2, +0.5, +1, +0.5,–2 for CS+, GS1, GS2, GS3, GS4, CS-). Note, that in contrast to the study of *McTeague et al., 2015*, generalization learning occurred along a continuum from CS+ to CS-, which is why the weight vector was adapted to only one half of the previously used model. For the quadratic and linear trends, the following weights were used, respectively: quadratic (weights: +2.5334, +1.0934,–0.0267, −0.8267,–1.3067, −1.4667) and linear (weights: +2.5, +1.5, +0.5,–0.5, −1.5,–2.5), paralleling previous work on fear generalization (*Ahrens et al., 2016*; *Lissek et al., 2014a*; *Lissek et al., 2014b*). The linear model analysis is mathematically insensitive to any linear transformation of the contrast vectors, which is why every vector was centered around zero to facilitate model comparisons. In each model, centered SPAI scores were entered as an additional predictor variable and subjects were entered as random intercepts to the model. Transitive Bayes factors (BFs) were then calculated for each candidate main effect, interaction and predictor weight model. Interpretation of Bayes factors follows guidelines developed by Jeffreys (1961). Bayesian analyses were conducted in R, using the package 'BayesFactor' (version 0.9) and default JZS-priors (*Rouder et al., 2012*). To follow up on the hierarchical model selection, individual visuo-cortical tuning indices were calculated for the lateral inhibition pattern. This index was defined as the scalar product of the weights of the lateral inhibition pattern and the subjects' individual ssVEP-response to the corresponding stimulus, so that higher/lower visuocortical tuning indices indicate stronger/weaker accentuation of the lateral inhibition pattern. Similar indices were calculated for the quadratic and linear trend. In a next step, these indices were correlated with the SPAI scores. For the frequentist analysis, Pearson's *r* was calculated, and the alpha level was set to .05, while the population correlation parameter $\rho$ was estimated for a two-sided Bayes factor analysis (*Ly et al., 2016*).

## Acknowledgements

This work was supported by the German Research Foundation – project number 44541416 - TRR-58, projects B05 of the 2nd funding period and B01 of the 3rd funding period

## Additional information

### Funding

| Funder | Grant reference number | Author |
|---|---|---|
| Deutsche Forschungsge-meinschaft | 44541416 - TRR-58 | Paul Pauli<br>Matthias J Wieser |

The funders had no role in study design, data collection and interpretation, or the decision to submit the work for publication.

### Author contributions

Yannik Stegmann, Data curation, Software, Formal analysis, Visualization, Writing - original draft, Writing - review and editing; Lea Ahrens, Conceptualization, Data curation, Investigation, Methodology; Paul Pauli, Conceptualization, Supervision, Funding acquisition, Visualization, Methodology, Project administration, Writing - review and editing; Andreas Keil, Data curation, Software, Supervision, Visualization, Methodology, Writing - review and editing; Matthias J Wieser, Conceptualization,

Resources, Data curation, Software, Formal analysis, Supervision, Funding acquisition, Methodology, Project administration, Writing - review and editing

## Author ORCIDs
Yannik Stegmann   https://orcid.org/0000-0002-0933-8492

## Ethics
Human subjects: Prior to participation, written informed consent was obtained from each participant. The study was approved by the ethics review board of the University of Würzburg (87/13).

## Decision letter and Author response
Decision letter https://doi.org/10.7554/eLife.55204.sa1
Author response https://doi.org/10.7554/eLife.55204.sa2

# Additional files

## Supplementary files
• Transparent reporting form

## Data availability
Data sets, the code for all analyses, as well as the code for the production of Tables 1-4, Figure 3, Figure 5 and Figure 7, are available at https://osf.io/4965f/.

The following dataset was generated:

| Author(s) | Year | Dataset title | Dataset URL | Database and Identifier |
|---|---|---|---|---|
| Stegmann Y, Ahrens L, Pauli P, Keil A, Wieser MJ | 2020 | Social aversive generalization learning sharpens the tuning of visuocortical neurons to facial identity cues | https://osf.io/4965f/ | Open Science Framework, 4965f |

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
