## [Decision Letter]

**Acceptance summary:**

Stegmann et al. address an important and interesting question, namely whether social fear conditioning leads to changes in neural processing of fear-conditioned faces versus other faces that are more or less similar to the acquired one. They investigate whether sharpened visuo-cortical tuning, a phenomenon usually reported for low-level visual features such as orientation, also affects the neural processing of social threat cues. The study uses a relatively simple and straightforward fear-conditioning paradigm and an non-invasive measure of brain electric activity (so-called steady-state visual evoked potentials, SSVEPs) to measure visual-cortical processing as well as a series of self-report measures. The results suggest sharpened tuning, correlated with interindividual differences in social anxiety. The reviewers and the Reviewing Editor collectively found considerable merit in the approach and findings and agreed that a case in terms of a lateral inhibition-like situation can be made, also in the light of earlier literature.

**Decision letter after peer review:**

Thank you for submitting your article "Social aversive generalization learning sharpens the tuning of visuocortical neurons to facial identity cues" for consideration by *eLife*. Your article has been reviewed by three peer reviewers, and the evaluation has been overseen by Jonas Obleser, the Reviewing Editor, and Richard Ivry as the Senior Editor. The following individuals involved in review of your submission have agreed to reveal their identity: Christian Keitel (Reviewer #1); Ulrike M. Krämer, PhD (Reviewer #3).

The reviewers and editors have discussed the reviews with one another, and the Reviewing Editor has drafted this decision to help you prepare a revised submission.

Summary:

The reviewers and the Reviewing Editor collectively found considerable merit in your approach and findings. We agreed that a case in terms of a lateral inhibition-like situation can be made, also in the light of earlier literature. However, there was clear consensus that the manuscript in its current form is not convincing yet and will require (1) refined statistical evidence being put forward, and (2) more detailed grounding in the principles of visual cortical processing and its modulation by (conditioned) fear, that is, a stronger framework behind the authors' assumptions and interpretations.

See below for further guidance to preparing a revised version.

Depending on revision and accompanying rebuttal letter, we will consult all or some of the present reviewers again.

Essential revisions:

– The degree to which the current data do show a sharpening of cortical tuning is contentious, at least. As such, title and Abstract may be misleading. SSVEPs are used, which measure population responses in visual cortex broadly (it is unclear whether the signals here are even stemming from face-specific areas – most likely not, as previous studies have shown that slower frequencies (<12Hz) are more effective in driving higher regions in visual cortex, see Baldauf and Desimone, 2014), and the "tuning" is inferred from a Bayesian model comparison that is rather indirect.

– While the study is trying to understand mechanisms of fear-conditioning, it lacks an account of how lateral inhibition in the processing of fear-conditioned faces would be implemented in the brain. The comparison to orientation tuning is challenging. Do the authors assume a face similarity map in visual cortex where similar faces are represented nearby each other, and so processing of similar faces can be attenuated through lateral inhibition? Would this happen via an attentional mechanisms, or through other connections? None of this seemed specified, and we struggled to understand of how such "tuning" would occur in fearful face representations.

– Related, the authors do not really come up with a suggestion how to link the sharpened tuning of the visuo-cortical responses on the one hand and the generalization of the subjective fear experience on the other hand. It is only stated that fear generalization is an "active and multifaceted process". As this is a central aspect of this study and pertains to the functional relevance of the observed enhanced tuning of the SSVEP response, one would like to see a bit more discussion of this discrepancy.

– For the generalization phase, the GS1-4 faces were treated as a single factor, which did not seem right to the reviewers. These should be separated into 4 different factors, given the authors' assumption that these 4 faces should show different modulations (i.e., lateral inhibition model and also linear decrease model).

– The support for lateral inhibition model only comes from a comparison of a series of different models to a random intercept model. This is weak evidence in favor of a lateral inhibition model. Before doing the modeling, it should first be tested whether the data for the GS1-4 faces differs reliably between each other (see last point), which is doubtful looking at the figure.

Second, a fair modeling approach would be to compare the models against each other (linear vs. lateral inhibition), and not take each model and compare it to the random intercept model, and then pick the model that shows the highest BF. Transitive BFs might be utilized here, accordingly.

– Each of the tested models does not appear to be well justified based on previous literature or any mechanistic accounts of fear conditioning, and it remains a bit unclear why these models were chosen.

– A potentially non-negligible difference (p = 0.07) between CS+ and CS- at baseline was noted, with the same directionality as the significant effect after fear condition. This seems problematic given that the difference remains relatively small after (and during) conditioning (though normalized effect sizes are not reported). We find such a small change extremely difficult to interpret. To be convinced that this is a real change in SNR through conditioning, we would want to see 1) statistics showing a reliable increase of the SSVEP response (so pre- and post training comparisons) which are currency lacking, and 2) effect sizes (at best with CIs for these, J.O.).

– The statistical results are in many places interpreted in a seeming all-or-none fashion, which does not seem appropriate. Effects with p-values of e.g. 0.05 are interpreted as true, while effects with p-values of e.g., 0.056 are interpreted as absent. Again, effect sizes should be reported, and a non-dichotomising interpretation of effects is encouraged.

– Related, the ANOVA results for the generalization show, if we understand correctly, a main effect for the CS-type but no main effect or interaction with social anxiety. The Bayesian linear models does find support for a full interaction model including a main effect and interaction with social anxiety. This discrepancy is not really resolved or discussed so far.

– Reviewers were not unequivocally convinced that the Current source density transforms are useful or adequate for SSVEPs. Please make sure to justify their use better (potentially also demonstrating their effect on overall conclusions/results, e.g. in a supplement). The authors should motivate their choice more strongly and report parameters λ/m.

– The authors have added an additional analysis that relates a measure of visuo-cortical tuning (i.e. the individual sharpening of response profiles after fear conditioning) to a measure of social anxiety. Indeed, they find a moderate positive correlation, strengthening their point that the individual sensitivity to threat cues can affect visuo-cortical tuning. Computing individual cortical tuning as weighted sum of SSVEP and model weights seems elegant. However, the reader is somewhat left at the authors' mercy to confide in the sensitivity of this measure. We strongly suggest to repeat this analysis with the weights of the Quadratic and Linear trend models and compare them with the lateral inhibition situation to show that there is no correlation in these cases.

– The manuscript is generally well-written and easy to read. The Results section however suffers from heavy in-text reporting of statistics. Reading flow will be improved by reporting stats in tables instead. Also consider the use of supplementary tables.

[Editors' note: further revisions were suggested prior to acceptance, as described below.]

Thank you for resubmitting your work entitled "Social aversive generalization learning sharpens the tuning of visuocortical neurons to facial identity cues" for further consideration by *eLife*. Your revised article has been evaluated by Richard Ivry (Senior Editor) and a Reviewing Editor.

The manuscript has been improved considerably.

Your work addresses an important and interesting question, namely whether social fear conditioning leads to changes in neural processing of fear-conditioned faces versus other faces that are more or less similar to the acquired one.

Reviewers and myself collectively found considerable merit in the approach and findings. We agreed that a case in terms of a lateral inhibition-like situation can be made, also in the light of earlier literature. The authors did a compelling job in carefully addressing most and certainly all major questions from us reviewers and editors.

We do believe there remain two issues that require further attention.

We anticipate that the reviewing and senior editor should be able to act on the revision without consulting the reviewers.

Requested revisions:

1) One reviewer alerted us that one issue raised in the initial round of reviews was not addressed, the potentially quite important pre-post comparison for SSVEP amplitude (i.e., pre-and post-training). In our view, it would be an important analysis to ask if there is an effect over time. We think proper reporting (and discussion) of such a pre-post comparison is important for making inferences that these results point to a change in neural tuning due to habituation.

2) The other remaining issue if of lesser concern:

You have not done a direct comparison of the models to each other. Please consider reporting these comparisons. The reviewer provides the following guidance: "While I follow that there is some benefit in comparing all models against a random intercept model, I would still want to see whether the differences between the GS1-4 faces differs reliably from each other (follow-up paired t-test)."

---

## [Author Response]

Essential revisions:– The degree to which the current data do show a sharpening of cortical tuning is contentious, at least. As such, title and Abstract may be misleading. SSVEPs are used, which measure population responses in visual cortex broadly (it is unclear whether the signals here are even stemming from face-specific areas – most likely not, as previous studies have shown that slower frequencies (<12Hz) are more effective in driving higher regions in visual cortex, see Baldauf and Desimone, 2014), and the "tuning" is inferred from a Bayesian model comparison that is rather indirect.

Defining what we mean by “tuning” is indeed crucial to avoid misconceptions. In the revised version, we give now an explicit definition of population-level tuning early on in the Introduction and explain the rationale for studying population-level tuning. Specifically, we base our conceptualization on conventions in vision neuroscience and psychophysics, where tuning functions are used to describe differences in sensitivity of a response (neural or behavioral) along a physical feature gradient. As the reviewers mention, probably best known is research conducted on orientation tuning of individual neurons in retinotopic visual cortex, meaning that visual neurons selectively respond to specific orientations as noted by the reviewers. There is also substantial evidence for single-unit and—importantly—population level (LFPs, fMRI) tuning in face-specific areas stemming from research in primates (e.g. Leopold et al., 2006; Freiwald et al., 2009; Freiwald and Tsao, 2010) and humans (e.g. Gilaie-Dotan et al. 2006; or Loffler et al., 2005). These studies have demonstrated that there are neurons and neuronal populations in face-sensitive cortical areas (patches in macaques), like the OFA and the FFA, which show gradual responses to varying facial identify, often referred to as “tuning” to facial identities. In this growing literature, authors often use the present approach, i.e. fitting a-priori tuning profiles to neural response data across a gradient of morphed faces (e.g., Koyano et al., 2019).

We agree with the reviewers that ssVEPs likely reflect population responses across different cortical areas. Changes of the tuning of a population response would then imply that the population response changes its profile across a feature gradient of interest. Sharpened tuning, as operationally defined in this sense then, is intended not to denote an absolute quality of the cortical tissue but the process in which the selective population response is altered across the gradient, in the present case by responding less to features that closely resemble a reference stimulus, or an attended stimulus. A shorter version of the above narrative is now included in the manuscript. This concern is also linked to concerns raised regarding the neural mechanism that would prompt population-level sharpening, addressed in the context of the next comment. We added the following section and a figure to illustrate different examples of tuning functions to our Introduction:

“The amplitude of the ssVEP differentiates threat from safety signals, being selectively heightened for conditioned threat cues (reviewed in Miskovic and Keil, 2012). […] Here, we examine the malleability of population-level tuning as observers learn to associate one identity along a gradient of morphs with an aversive outcome.”

Finally, to adress the concerns regarding slower frequencies being more effective in eliciting signals in face-specific areas, we added a paragraph with references to the Materials and methods section, stating that even though face-specific areas are often targeted with relatively slow driving frequencies (Baldauf and Desimone, 2014), face-specific processing has also been isolated from paradigms with faster frequencies, when combined with appropriate experimental manipulations (Campagnoli et al., 2019; Gruss, Wieser, Schweinberger and Keil, 2012; Wieser, Flaisch and Pauli, 2014; Wieser et al., 2014)

– While the study is trying to understand mechanisms of fear-conditioning, it lacks an account of how lateral inhibition in the processing of fear-conditioned faces would be implemented in the brain. The comparison to orientation tuning is challenging. Do the authors assume a face similarity map in visual cortex where similar faces are represented nearby each other, and so processing of similar faces can be attenuated through lateral inhibition? Would this happen via an attentional mechanisms, or through other connections? None of this seemed specified, and we struggled to understand of how such "tuning" would occur in fearful face representations.

We expanded on the hypothetical neural mechanisms in the revised version of the manuscript. In brief, based on the animal studies cited above, and based on the extant work on neuroplastic changes for tuning to low level features, we considered two hypotheses that rely on the same computational principle: As stated in McTeague et al., 2015 in the case of orientation tuning, signals from frontoparietal attention networks may selectively faciliate CS+ representations in visual cortex, prompting local inhibitory interactions between adjacent cortical units (here: orientation columns). This process is thought to prompt suppression of the features represented by the most spatially proximal populations. We have developed a computational model (currently under review) that explains these and other data best by assuming that top-down signals take the shape of a generalization gradient (paralleling behavioral and autonomic data), and it is the organization of visual cortex that turns this gradient into a sharpening pattern through lateral inhibition. As such, sharpening would not be inherited by anterior areas but would result from the organization of visual cortical areas.

In the case of facial identity gradients, as alluded to by the reviewers, there are two loci where such tuning may occur, each of which being supported by published findigs:

First, as aversive learning proceeds, projections from anterior structures may increasingly target lower-level representations of individual facial features in retinotopic areas, thus prompting inhibitory interactions between individual physical facial features such as orientation or spatial frequency, which differ across the morphing gradient. This is consistent with findings and models in perceptual learning (e.g. reverse hierarchy theory; Ahissar and Hochstein, 2004) and would prompt a mid-occipital topography of sharpening effects in the present study.

Second, inhibitory interactions may occur between similar faces along a gradient of morphs, in face-sensitive cortical areas. This alternative hypothesis is in line with recent evidence suggesting that neuronal populations in face sensitive cortical patches encode identity by combining their population tuning to sets of high-order shape and appearance dimensions (e.g., Chang and Tsao, 2017). In the present study, this hypoythesis would be supported by a right-occipital topography of sharpened tuning.

The full discussion of the neural mechanisms now reads as follows:

“McTeague et al., 2015, using a generalization gradient of oriented gratings, explained their finding of visuocortical sharpening as a consequence of lateral inhibitory interactions among orientation-selective neuronal populations in the primary visual cortex. […] Thus, our results are compatible with the idea of a right-hemispheric dominance of face perception (Rossion, 2014) and suggest an involvement of the right OFC or FFA in sharpening face discrimination on the basis of lateral inhibition.”

– Related, the authors do not really come up with a suggestion how to link the sharpened tuning of the visuo-cortical responses on the one hand and the generalization of the subjective fear experience on the other hand. It is only stated that fear generalization is an "active and multifaceted process". As this is a central aspect of this study and pertains to the functional relevance of the observed enhanced tuning of the SSVEP response, one would like to see a bit more discussion of this discrepancy.

This is indeed a central aspect of our study. We hope that the account above assisted in making explicit the hypothesis that generalization occurs widely in brain and behavior, as established in decades of research, and that sensory areas turn generalization-tuned top-down signals into local sharpening because of their organizational principle of lateral inhibition. This hypothesis is now made more explit in the manuscript. In addition, we extended the paragraph, where we thoroughly discuss the discrepancies between sensory and efferent systems from an evolutionary perspective, by a discussion on the underlying mechanisms of the different output systems based on the existing literature. It reads as follow:

“A large body of work has shown that visuocortical engagement with specific stimulus features varies with the motivational relevance of these features (Bradley, Keil and Lang, 2012). […] This evolutionary interpretation assumes that the constantly changing and diverse environment in which humans find themselves demands and thus favors plastic physiological mechanisms, which may differ between systems in order to optimize functioning (Miskovic and Keil, 2012).”

– For the generalization phase, the GS1-4 faces were treated as a single factor, which did not seem right to the reviewers. These should be separated into 4 different factors, given the authors' assumption that these 4 faces should show different modulations (i.e., lateral inhibition model and also linear decrease model).

Thank you, for pointing out this potential source of confusion. We completely agree that the different generalization stimuli represent separate levels of the analysis, and had entered them accordingly in the previous version. However, to avoid further confusion, we changed the description of the factor on page x from “CS-type (6: CS+ *vs* GS1-4 *vs* CS-)“ to “CS-type (6 levels: CS+ *vs* GS1 vs GS2 vs GS3 vs GS4 *vs* CS-)”.

– The support for lateral inhibition model only comes from a comparison of a series of different models to a random intercept model. This is weak evidence in favor of a lateral inhibition model. Before doing the modeling, it should first be tested whether the data for the GS1-4 faces differs reliably between each other (see last point), which is doubtful looking at the figure.Second, a fair modeling approach would be to compare the models against each other (linear vs. lateral inhibition), and not take each model and compare it to the random intercept model, and then pick the model that shows the highest BF. Transitive BFs might be utilized here, accordingly.

Again, we agree that comparisons to the random intercept (Null model), would have resulted in weak evidence, only. However, by comparing every model against the same random intercept model, the resulting BFs share a common denominator. Thus, as mentioned by the reviewers, this allows us to use transitive BFs, which we used in Table 4 (and in the text), and which we now emphasize in the revised version throughout. This is an important prerequisite for consecutive model comparisons. Here, we not only compared models in which we hierarchically included the main and interaction effects of CS-type and social anxiety for each contrast model by dividing the respective BFs, but we also compared the contrast models against each other. The results of these comparisons can be seen in the last two columns of Table 4, showing that the lateral inhibition model is 858 times more likely than the quadratic trend model and 502 times more likely than the linear trend model. To clarify this, we added the following paragraph to the Results section:

“Importantly, by comparing the main and interaction effect models of each contrast against the same random intercept model, it was possible to derive the relative evidence of the lateral inhibition model over the quadratic and linear trend (see Table 4, last two columns). Results for the full interaction model demonstrated that the lateral inhibition model is 858 times more likely than the quadratic trend model and 502 times more likely than the linear trend model.”

– Each of the tested models does not appear to be well justified based on previous literature or any mechanistic accounts of fear conditioning, and it remains a bit unclear why these models were chosen.

We expanded on the background of the models in the Introduction and Discussion. In our study, we compared three different models; the lateral inhibition model, a linear and a quadratic trend model. We decided to use the linear and quadratic trend as comparisons since they are frequently employed in the fear generalization literature, where they are used to indicate an overgeneralization of fear (e.g. Lissek, 2014; Lissek, 2014; Ahrens, 2016). In contrast to the trend analyses in these studies, we explicitly expressed these trends as numeric values, which is mathematically equivalent to the implicit approach. The contrast vectors are also insensitive to any linear transformations, as long as the linear and quadratic properties stay intact. Our main model, the lateral inhibition model was based on previous studies on visuocortical tuning (McTeague, 2015; and Antov et al., 2020), which expressed the lateral inhibition model as a difference of two Gaussian distributions. However, as we have stated in our Materials and methods section, we adapted the lateral inhibition model to only one half of the previously used model, as in our study, generalization learning occurred along a continuum from the threat stimulus (CS+) to the safety stimulus (CS-) and not along a continuum with the CS+ is in the middle. To facilitate model comparisons, the sum of each contrast vectors was rescaled to equal zero. We included the following description with references to the Results section:

“The lateral inhibition pattern was expressed as the difference of two Gaussians (weights: +2, -2, +0.5, +1, +0.5, -2 for CS+, GS1, GS2, GS3, GS4, CS-), paralleling previous studies on visuocortical tuning (Antov et al., 2020; McTeague et al., 2015). The quadratic (weights: +2.5334, +1.0934, -0.0267, -0.8267, -1.3067, -1.4667) and linear (weights: +2.5, +1.5, +0.5, -0.5, -1.5, -2.5) trend were modelled after the analyses of the linear and quadratic component of the generalization gradient, which are commonly employed in the fear generalization literature (Ahrens et al., 2016; Lissek et al., 2014; Lissek et al., 2014).”

And updated the Materials and methods section to:

“To compare the lateral inhibition pattern to a quadratic and linear fit, Bayesian linear models were used. […] The linear model analysis is mathematically insensitive to any linear transformation of the contrast vectors, which is why every vector was centered around zero to facilitate model comparisons.“

– A potentially non-negligible difference (p = 0.07) between CS+ and CS- at baseline was noted, with the same directionality as the significant effect after fear condition. This seems problematic given that the difference remains relatively small after (and during) conditioning (though normalized effect sizes are not reported). We find such a small change extremely difficult to interpret. To be convinced that this is a real change in SNR through conditioning, we would want to see 1) statistics showing a reliable increase of the SSVEP response (so pre- and post training comparisons) which are currency lacking, and 2) effect sizes (at best with CIs for these, J.O.).

You address an important point, which we indeed did not consider earlier. There seem to be two concerns here:

First, do ssVEP amplitudes change from habituation to acquisition? To test for pre and post-training differences, we directly compared the difference between CS+ and CS- in ssVEP-SNRs during habituation versus acquisition as well as during habituation versus generalization with simple t-tests. However, the t-test for CS-differences in habituation compared to acquisition, t(66) = 0.83, p = .409, d = 0.10, CI = [-0.14; 0.34], as well was the t-test for habituation compared to generalization, t(66) = 0.52, p = .606, d = 0.06, CI = [-0.18; 0.30] showed no significant differences. This is to a certain extent a limitation to our study, which we now address in our Discussion. Yet, in the differential conditioning literature, cross-phase comparisons are typically avoided because they confound any conditioning effects with time-dependent effects such as adaptation, in which both the CS+ and CS- evoked responses decline over time, from habituation to acquisition and finally into the extinction phase. This has been well established in meta-analyses (e.g., Fullana et al., 2016), and is true for a wide range of dependent measures, including startle, skin conductance, fMRI, and ssVEPs where this pattern of findings has been noted and systematically addressed several times (e.g., Moratti et al., 2005; Keil et al. 2013). Thus, fear conditioning studies interested in temporal changes previously utilized analysis on a trial-by-trial level in order to avoid confoundation with these types of adaptation processes (e.g. Sjouwerman, Niehaus and Lonsdorf, 2015; Weike, Schupp and Hamm, 2007; Wieser, Miskovic, Rausch and Keil, 2014).

Second, is there a benchmark conditioning effect, given that there are visible differences in the same direction during habituation? To address this, we conducted a post-hoc analysis quantifying the amount of conditioning-related effects above and beyond the initial difference in habituation by means of parametric bootstrapping in combination with Bayesian analyses (Efron, 2012). We computed a distribution for the CS+ versus CS- difference in ssVEP amplitude separately for habituation (to be used as prior distribution) and acqusition (posterior distribution) based on 100000 bootstrapped group mean differences. Odds for conditioning effects to occur were estimated from these two distributions (i.e., the odds of the CS+-CS- difference to be positive), and the BF of interest is given as the ratio of posterior (acquisition) over prior (habituation) odds. The error of this process was estimated by repeating the above process 100 times and measuring the standard error of the resulting mean. This BF corresponds to the change in confidence that the ssVEP for the CS+ is greater than ssVEP for the CS- in acquisition (the posterior), relative to the prior distribution, which was estimated from the CS+ versus CS- difference during habituation. This analysis yielded a BF (posterior odds over prior odds) of 4.77, error=1.3%, suggesting that the acquisition data provided substantial evidence for the notion that fear conditioning selectively amplifies the ssVEP, above and beyond the differences present during habituation. This is now mentioned in the Discussion. Finally, when using face CSs, differences during habituation often indicate different responses to one of the two faces, an explanation which seems unlikely in our study, since the mapping of the two faces to the CS+ and CS- was fully counterbalanced between subjects.

In conclusion, the present data provide evidence for the notion that conditioning selectively heightens the ssVEP evoked by the CS+. This being said, we also emphasize in the manuscript that a lack of significant changes between the experimental phases, which were kept short to allow ample generalization learning, would not affect our interpretation of the main results regarding the generalization phase. We added the following qualifications to our Discussion:

“One important limitation to our findings is that there was no substantial increase in differential ssVEP-SNRs (CS+ vs CS-) from habituation to acquisition, t(66) = 0.83, p = .409, d = 0.10, CI95 = [-0.14, 0.34]. […] However, even though there were reliable differences between CS+ and CS- during acquisition and generalization, future studies may replicate our findings using a different set of stimuli to rule out the influence of baseline differences.”

– The statistical results are in many places interpreted in a seeming all-or-none fashion, which does not seem appropriate. Effects with p-values of e.g. 0.05 are interpreted as true, while effects with p-values of e.g., 0.056 are interpreted as absent. Again, effect sizes should be reported, and a non-dichotomising interpretation of effects is encouraged.

We agree on the importance of effect sizes and their confidence intervals, which is why we already reported R²-values and their 95%-Cis as effect sizes for the discrete factors (CS-type) and betas and their SE as effect sizes for the continuous factors (social anxiety) in the linear mixed model analyses (but please also note this discussion on the issues of effect sizes in linear mixed model analyses: https://stats.stackexchange.com/questions/95054/how-to-get-an-overall-p-value-and-effect-size-for-a-categorical-factor-in-a-mi). To further discourage a dichotomizing interpretation of effects, we rephrased the Results section carefully.

– Related, the ANOVA results for the generalization show, if we understand correctly, a main effect for the CS-type but no main effect or interaction with social anxiety. The Bayesian linear models does find support for a full interaction model including a main effect and interaction with social anxiety. This discrepancy is not really resolved or discussed so far.

This is true and, importantly, the effect of social anxiety received only support for the lateral inhibition model. This discrepancy is not a conflict between the ANOVA and Bayesian-analysis but results from the differences between a non-directional omnibus-test (which the ANOVA is) and the explicit test of specific contrasts (which we did with Bayesian-analyses). While the omnibus-analysis tests the influence of social anxiety in any direction, the contrast analyses tests if social anxiety is associated with responses following the specified pattern only, which strongly increases the statistical power. In line with this, it is important to mention, that social anxiety was not associated with the quadratic or linear trend model. To clarify this matter, we added following to the Discussion:

“Please note that the effect of social anxiety on visuocortical responses was only evident in the Bayesian-analysis but not in the corresponding ANOVA. This discrepancy results from the differences regarding statistical power between omnibus- and contrast-analysis (Furr and Rosenthal, 2003). The ANOVA tests for any differences, while the contrast-analysis only tests for deviations from the specified pattern. This notion is substantiated by the finding that social anxiety was not associated with visuocortical responses for the quadratic or linear trend model, but only for the lateral inhibition model.”

– Reviewers were not unequivocally convinced that the Current source density transforms are useful or adequate for SSVEPs. Please make sure to justify their use better (potentially also demonstrating their effect on overall conclusions/results, e.g. in a supplement). The authors should motivate their choice more strongly and report parameters λ/m.

We have added a paragraph highlighing the benefits of the CSD transformation for reducing unwanted topographical variability between subjects by quantifying source densities. We cite methodological papers and empirical papers using this technique for that purpose. We also report now that we used a λ = .2 for all analyses. The paragraph in the Materials and methods section reads as follows:

“To increase spatial resolution of the EEG signal, we then calculated the current source densities (CSD) of the time-averaged data. The CSD transformation offers the advantage of reducing the negative impact of volume conduction and thereby effectively minimizing unwanted topographical variability between subjects by quantifying source densities (Junghöfer et al., 1997; Kayser and Tenke, 2015). The CSD approach relies on the spatial Laplacian (the second spatial derivative) of the scalp potential to estimate the potential distribution at the cortical surface. Here we used the CSD algorithm described by Junghöfer et al., 1997, with λ = .2, which is well suited for dense-array EEG montages and has been used in previous studies investigating ssVEP responses to facial stimuli (McTeague et al., 2011) or during fear generalization (McTeague et al., 2015).”

– The authors have added an additional analysis that relates a measure of visuo-cortical tuning (i.e. the individual sharpening of response profiles after fear conditioning) to a measure of social anxiety. Indeed, they find a moderate positive correlation, strengthening their point that the individual sensitivity to threat cues can affect visuo-cortical tuning. Computing individual cortical tuning as weighted sum of SSVEP and model weights seems elegant. However, the reader is somewhat left at the authors' mercy to confide in the sensitivity of this measure. We strongly suggest to repeat this analysis with the weights of the Quadratic and Linear trend models and compare them with the lateral inhibition situation to show that there is no correlation in these cases.

We would like to thank the reviewers for this suggestion. We now have calculated the scalar product for the quadratic and linear trend models and added the corresponding regression analyses to the Results section. We also updated the figure accordingly. The text now reads as follows:

“In a first step, we calculated a visuocortical tuning index on subject level, which is the scalar product of the weights of the lateral inhibition model (2, -2, 0.5, 1, 0.5, -2) and the respective individual ssVEP responses. […] The regression analysis revealed a moderate, positive correlation between social anxiety and the visuocortical tuning index, *r*(65) = .288, *p* = .018, *BF_1/0_* = 3.65, confirming that the accentuation of the lateral inhibition pattern increased with higher social anxiety, while the weak correlations between social anxiety and the index for the quadratic, *r*(65) = .146, *p* = .238, *BF_2/0_* = 0.52, or linear trend, *r*(65) = .137, *p* = .268, *BF_3/0_* = 0.49, did not reach the frequentist significance threshold and did not receive support from Bayesian analyses.”

– The manuscript is generally well-written and easy to read. The Results section however suffers from heavy in-text reporting of statistics. Reading flow will be improved by reporting stats in tables instead. Also consider the use of supplementary tables.

We agree and have moved the statistics of all omnibus tests from the text to tables. We accordingly revised the Results section.

[Editors' note: further revisions were suggested prior to acceptance, as described below.]

Requested revisions:1) One reviewer alerted us that one issue raised in the initial round of reviews was not addressed, the potentially quite important pre-post comparison for SSVEP amplitude (i.e., pre-and post-training). In our view, it would be an important analysis to ask if there is an effect over time. We think proper reporting (and discussion) of such a pre-post comparison is important for making inferences that these results point to a change in neural tuning due to habituation.

Thank you for bringing this issue to our attention. We can only assume that we failed to explain and appropriately highlight the pre-post comparisons that we included in the revised version (t-tests and a directed Bayesian bootstrapped test that directly tests the conditioning effects on SSVEP amplitude pre- versus post-conditioning). To avoid confusion, in the last response letter, we reported and discussed pre-post training comparisons analyzed with t-tests (including effect sizes and CIs) and conducted an additional bootstrap-based Bayesian test to examine conditioning-related changes above and beyond the initial difference in habituation. As a reminder, the BF for a pre-post conditioning effect (CS+ minus CS-) was 4.7, but some of the direct amplitude comparisons of the same conditions from pre to post, did not reach the level of NHST significance, with small effect sizes, prompting us to add two paragraphs of discussing this as a limitation. To make this comparison more obvious and make sure that readers will find this important information in the manuscript, we added a subheading, and added language that highlights these tests.

2) The other remaining issue if of lesser concern:You have not done a direct comparison of the models to each other. Please consider reporting these comparisons. The reviewer provides the following guidance: "While I follow that there is some benefit in comparing all models against a random intercept model, I would still want to see whether the differences between the GS1-4 faces differs reliably from each other (follow-up paired t-test)."

This comment seems to raise two different issues. First, how does the lateral inhibition model perform against the quadratic or linear trend model? This being an important point, we already had addressed this question in the original manuscript by not only comparing each model against the random intercept model, but we also directly compared the different models to each other. We also elaborated on this approach in the initial round of revisions (see Comment #5). Again, we can only assume that this information is not obvious enough in its presentation, so we amplified its visibility by adding additional sentences explaining and highlighting them.

Second, we indeed did not report follow-up contrasts for ssVEP amplitudes, since we addressed the follow-up analysis with the more powerful, and in our view more elegant and theory-driven, model comparisons. In response to the reviewer’s concern and, to keep reporting of the ssVEP results consistent with the results for ratings, we now report post-hoc contrasts (t-tests, as requested) for the linear mixed model analysis of ssVEP amplitudes. In line with the fear generalization literature, all contrasts were referenced to the conditioned safety cue (Lissek et al., 2008; Lissek et al., 2010). Please note that we are not able to calculate standardized effect sizes for contrasts in linear mixed model analysis. We added the following paragraph to the ssVEP Results section:

“Post-hoc contrasts indicated significant differences between GS4 *vs* CS-, *t*(325)=2.94, *p*=.003, GS3 *vs* CS-, *t*(325)=3.24, *p*=.001, and CS+ *vs* CS-, *t*(325)=2.17, *p*=.031, but not between GS2 *vs* CS-, *t*(325)=1.75, *p*=.081, and GS1 *vs* CS-, *t*(325)=0.82, *p*=.411.”